# [Re]Reproducibility Study: Cluster-guided Contrastive Graph Clustering Network

## Abstract

Contrastive learning has shown promise in deep graph clustering, but current methods have notable limitations. Positive sample effectiveness is heavily influenced by data augmentation, risking semantic drift and improper pair selection. Moreover, negative pairs lack reliability, disregarding crucial clustering information. To address these issues, the authors proposed Cluster-guided Contrastive Graph Clustering Network (CCGCN). It employs a unique Siamese encoder architecture, creating two distinct graph views without complex augmentations, enhancing positive sample discriminative quality. Negative samples are selected from cluster centers for improved reliability. An objective function encourages intra-cluster cohesion and inter-cluster separation. Initially evaluated on six datasets, we expand to 12 graph and 2 non-graph datasets in our study, aiming to validate and generalize the method's effectiveness through reproducibility and additional experiments [1]

## 1 Reproducibility Summary

### 1.1 Scope of Reproducibility

This research focuses on assessing the reproducibility of the paper titled "Cluster-guided Contrastive Graph Clustering Network" [(Yang et al., 2023)] with the aim of validating the claims made in the paper. The specific assertions being investigated are as follows (1)Proposal of a Discriminative Positive Sample Extraction Mechanism (DPS), the paper claims the introduction of a positive sample extraction mechanism (DPS) that is more discriminative. (2) Proposal of unshared Siamese encoder to prevent Semantic Drift, the paper asserts the proposal of an unshared Siamese encoder as a strategy to prevent inappropriate graph augmentation that could lead to semantic drift. (3)Proposal of an algorithm for constructing Reliable Negative Samples (RNS), the paper puts forth a claim about a proposed algorithm for constructing more reliable negative samples (RNS).

### 1.2 Methodology

The authors shared their PyTorch code, which was subsequently modified for two primary purposes: first, to expand the experiments to include additional graph & non graph datasets, and second, to perform the ablation study. All experiments, including both the original and extended ones, were conducted on Google Colab, utilizing the T4 GPU for computational acceleration. The total time required for the entire set of experiments amounted to approximately 20 hrs.

### 1.3 Results

The study achieved comparable results to the original paper when evaluating performance on existing datasets. However, in the ablation study, although the results did not precisely match to those obtained by the authors, they followed the same trend. Furthermore, we extended the experiments to include additional

---

[1]Our GitHub : `https://anonymous.4open.science/r/Rep_CCGC-BB93/README.md`

graph & non graph datasets and assessed the model's generalizability and performance by performing hyperparameter sensitivity and cross validation studies. We have discussed in detail the problems and solution that are faced when extending to non graph datasets.

### 1.4 What was Easy

The approach outlined in the paper was presented in a clear and concise manner, facilitating the straightforward implementation of the provided code. Despite the GitHub repository from the authors containing code for only one dataset, the implementation process was easily extended to accommodate additional datasets. Minor adjustments to the code were sufficient for this extension.

### 1.5 What was difficult

Recreating some of the experiments in the ablation study posed challenges, as it necessitated the replacement of portions of the existing code with components from other papers. We also faced difficulties while extending the paper by applying the model to non graph datasets, due to computational restraints.

### 1.6 Communication with the original authors

Efforts were made to reach out to the authors of the original paper. However, despite these attempts, no response was received from the authors.

## 2 Introduction

Contrastive learning has demonstrated notable success in the realm of deep graph clustering by leveraging constructed pairs to unveil the requisite feature distribution for effective clustering. Despite its effectiveness, existing methods exhibit two notable drawbacks.

- The effectiveness of positive samples in contrastive learning for deep graph clustering is strongly influenced by the choice of data augmentation methods. Inappropriately applied data augmentation can result in semantic drift [Lee et al. (2021)] and lead to improper selection of positive pairs.

- The negative pairs created are not sufficiently reliable to disregard important clustering information.

The authors proposed Cluster-guided Contrastive Graph Clustering Network (CCGCN) employs a unique Siamese encoder with unshared weights between the sibling networks. This design creates two distinct views of the graph without resorting to complex graph augmentations like node or edge perturbations, thereby avoiding potential issues of semantic shift. By leveraging high-confidence clustering information, the model selects positive pairs from the same clusters in the two views, enhancing the discriminative quality of positive samples. Additionally, negative samples are chosen from the centers of other clusters, contributing to increased reliability in negative sample selection.

Finally, an objective function is employed to bring samples from the same cluster closer together while simultaneously pushing away samples from different clusters. The efficacy of this approach was initially evaluated on six datasets in the original paper. In our study, we have extended this comparison to a total of 12 graph and 2 non graph datasets, broadening the scope of the assessment.

In this reproducibility study we aim to validate the authors claim on the effectiveness of their proposed method by reproducing the authors results, and to provide insights to the genralizability of their approach by performing additional experiments.

## 3 Scope of Reproducibility

The authors present a novel deep graph clustering approach that employs contrastive learning. In their work, they make the following proposals,

- **Discriminative Positive Sample Extraction Mechanism (DPS):**

  – Claim: DPS is more discriminative compared to traditional positive sample extraction mechanisms.
  – Implication: The proposed DPS method enhances the quality of positive samples, leading to improved performance in graph clustering tasks.

- **Unshared Siamese Network for Embedding:**

  – Claim: Creating two embeddings via an unshared Siamese network is superior to traditional data augmentation methods and helps avoid semantic drift.
  – Implication: The use of unshared Siamese networks for embedding prevents semantic drift, offering advantages over conventional data augmentation techniques and resulting in more effective graph clustering.

- **Reliable Negative Samples (RNS):**

  – Claim: The proposed method of Reliable Negative Samples constructs more reliable negative samples, contributing significantly to the learning process.
  – Implication: Generating reliable negative samples enhances the informativeness of the training data, improving the model's ability to distinguish between different graph clusters.

To substantiate their claims, we conducted two primary experiments.

1. **Experiment 1 : Performance comparison**
   The performance of CCGCN was compared over 6 datasets and the reproduced results were compared to that obtained by the authors. The evaluation of the methods was based on four metrics namely accuracy, Normalized Mutual Information, Adjusted Rand Index and F1.

2. **Experiment 2 : Ablation study**
   In the second experiment, we conducted an ablation study to evaluate the effectiveness of each component in their proposed approach. The study involved systematically replacing parts of their existing method with standard methods. Since the authors did not share the code and details for this experiment, we independently implemented and tried to reproduce the ablation study.

The following section discusses the methodology employed in this study to validate the paper. This includes details about reproducing Experiment 2, the adjustments made to the code and the rationale behind extending the evaluation to additional datasets. Through these efforts, we aim to validate the robustness and applicability of the proposed approach, contributing to a more comprehensive understanding of its performance.

## 4    Methodology

To reproduce the results in experiment 1 we relied on the code provided by them, we reproduced experiment 1, which demonstrated the enhanced efficacy of CCGCN in comparison to alternative models, the detailed code for Experiment 1 was obtained from their GitHub repository[2].
To validate the claims we primarily performed an ablation study, through which we discussed the effectiveness and contributions of DPS , RNS & the unshared Siamese encoder. Although the authors didn't share the code for Experiment 2, we managed to replicate a substantial part of it by modifying and expanding the codebase and leveraging the resources cited in the original paper.
Additionally to the original paper, we applied this model to non graph datasets, to assess its performance and robustness on different type of data.

---

[2]GitHub repository of the authors: `https://github.com/xihongyang1999/CCGC/tree/main`

### 4.1 Model Description

#### 4.1.1 Distinct sample view construction

In the proposed architecture in fig 1, we see that the authors are inspired by SCGC Liu et al. (2022) adopted a similar approach, by embedding nodes into the latent space. To achieve this, they constructed two distinct sample views through the implementation of parameter unshared Siamese encoders

Prior to encoding, neighbour information aggregation is performed by using Laplacian filter Cui et al. (2020), this filtering process is essential for capturing the local and global structure of the graph.

Table 1: List of Notations

| Notation | Description |
|---|---|
| $G = \{X, A\}$ | Undirected graph |
| $N$ | Number of nodes |
| $K$ | Number of classes |
| $V = \{v_1, v_2, \ldots, v_N\}$ | Set of nodes |
| $\epsilon$ | Set of edges |
| $X \in \mathbb{R}^{N \times D}$ | Attribute matrix |
| $A \in \mathbb{R}^{N \times D}$ | Adjacency matrix |
| $d_i = \sum_{(v_i, v_j) \in \epsilon} a_{ij}$ | Degree |
| $D = \mathrm{diag}(d_1, d_2, \ldots, d_N) \in \mathbb{R}^{N \times N}$ | Degree matrix |
| $L = I - \bar{D}^{-1/2} \bar{A} \bar{D}^{1/2}$ | Symmetric normalized graph Laplacian matrix |

$$\bar{X} = (I - \bar{L})^T \cdot X \tag{1}$$

$\bar{X}$ is a representation of the attribute feature after a smoothing process. Let $t$ denote the number of layers in the Laplacian filter.The encoding of $\bar{X}$ is carried out using unshared Siamese encoders. The output of this encoding process results in two distinct views, represented as $E^{v1}$ and $E^{v2}$. These views capture different representations of the input $\bar{X}$.

$$E^{v1} = \text{Encoder}_1(\bar{X}), E^{v2} = \text{Encoder}_2(\bar{X}). \tag{2}$$

The encoders are designed to have the same architecture but unshared learnable parameters thus avoiding semantic drift. Subsequently $E^{v1}$ and $E^{v2}$ are normalised using $l^2$ norm.

$$E^{v2} = \frac{E^{v1}}{\|E^{v2}\|_2} E^{v2} = \frac{E^{v1}}{\|E^{v2}\|_2} \tag{3}$$

#### 4.1.2 Pseudo label construction

The authors propose cluster guided contrastive learning, i.e, clustering by mining the high-confidence clustering information. Views of the 2 embeddings are fused together and K means clustering is performed on the fused view.

$$E = \frac{E^{v1} + E^{v2}}{2}$$
$$CONF_i = e^{\|E_i - C_p\|_2} \tag{4}$$

$C_p\{p = 1, 2...\}$ denotes the center of the high confidence cluster which contains the $i^{th}$ sample. Thus high confidence index are calculated where $h_i$ denotes the $h_i^{th}$ sample belonging to the top $\tau$ high confidence sample set.

#### 4.1.3 Discriminative Positive Sample Construction Strategy & Reliable Negative Sample Construction Strategy

DPS was designed to increase the discriminitave capabilities of the model, corresponding nodes of high confidence indices are selected in the two views. the selected nodes are grouped into $K$ disjoint clusters in their respective views ie, $B_p^{v_1}\{p = 1, 2..\}$ & $B_q^{v_2}\{q = 1, 2..\}$. The constructed clusters are the "high confidence clusters" with pseudo label, these are used to further construct positive samples. Thus, pseudo labels are

utilized.

The centers of the high confidence cluster are calculated and these are used to create the negative samples, this method increases the reliability of the created negative samples.

### 4.1.4 Objective loss Functions

The proposed objective loss function contains two part namely, positive loss and negative loss. The positive loss is constructed between the normalized cross-view positive sample embeddings

$$L_{\text{pos}} = \frac{1}{K} \sum_{p=1}^{K} \sum_{i=1}^{n_p} \|B_{p[i,:]}^{v1} - B_{p[i,:]}^{v2}\|_2 \tag{5}$$

$B_{p[i,:]}^{v1}$ & $B_{p[i,:]}^{v2}$ denotes i-th normalized node embedding in the p-th cluster of the first and second view & $n_p$ is number of high-confidence samples in the p-th cluster

Negative loss is the cosine similarity between different centers of the high confidence embeddings, here $C_p^{v1}$ is is the p-th high-confidence center in the first view.

$$L_{\text{neg}} = \frac{1}{K^2 - K} \sum_{p=1}^{K} \sum_{q=1}^{k} \frac{C_p^{v1}, C_q^{v2}}{\|C_p^{v1}\|_2 \cdot \|C_q^{v2}\|_2} \tag{6}$$

The total loss formulated is,

$$L = L_{pos} + \alpha \cdot L_{neg} \tag{7}$$

$\alpha$ is the trade off between positive and negative loss.

### 4.2 Datasets

Table 2: Dataset Information

| Dataset | Samples | Dimension | Edges | Classes |
|---------|---------|-----------|-------|---------|
| Presented in paper: | | | | |
| CORA | 2708 | 1433 | 5278 | 7 |
| CITESEER | 3327 | 3703 | 4552 | 6 |
| BAT | 131 | 81 | 1038 | 4 |
| EAT | 399 | 203 | 5994 | 4 |
| UAT | 1190 | 239 | 13599 | 4 |
| AMAP | 7650 | 745 | 119081 | 8 |
| Extended: | | | | |
| ACM | 3025 | 1870 | 13128 | 3 |
| DBLP | 4057 | 334 | 3528 | 4 |
| AMAC | 13752 | 767 | 24586 | 10 |
| TEXAS | 183 | 1703 | 162 | 5 |
| WISC | 251 | 1703 | 257 | 5 |
| PUBMED | 19717 | 500 | 44324 | 3 |

Table 2 presents the statistical summary of the datasets[3] used in the study. The code provided by the authors was designed for a single dataset, employing files in the '.allx', '.ally' & '.graph' formats—all of which are pickled NumPy files. However, the datasets acquired from the internet had a different file format, specifically in ".npy" format.

Consequently, we developed our own data loader function to accommodate these datasets with the ".npy" format, addressing the variation in data file formats across different sources.

In addition to these traditional graph datasets, we were also able to generalize and assess the performance on non - graph datasets such as USPS & REUT, though we encountered numerous challenges in their application, which we have discussed in section 6.0.1.

Table 3: Non graph dataset Information

| Dataset | Samples | Dimension | Type | Classes |
|---------|---------|-----------|------|---------|
| USPS | 9298 | 256 | Image | 10 |
| REUT | 10000 | 2000 | Text | 4 |

### 4.3 Hyperparameters

The two major hyperparameters which affected the results were $\alpha$ and $\tau$. $\alpha$ controls the trade-off between positive and negative losses. Through our study, we agree with the author that the variation of $\alpha$ generally

---

[3]data were obtained from `https://github.com/yueliu1999/Awesome-Deep-Graph-Clustering`

has minor impact on the final results as compared to $\tau$.

Table 4: Hyperparameter Selection after grid search

| Dataset | $\alpha$ | $\tau$ |
|---------|----------|--------|
| CORA | 1 | 0.7 |
| CITESEER | 0.5 | 0.7 |
| AMAP | 0.1 | 0.65 |
| UAT | 0.5 | 0.65 |
| EAT | 10 | 0.65 |
| BAT | 10 | 0.7 |

The parameter $\tau$ serves as a critical threshold in the analysis, delineating the 'top $\tau$ high-confidence sample set' where samples with confidence levels surpassing this cutoff are included in the high-confidence clustering index.

In the paper, authors have mentioned that when $\tau < 50\%$, the discriminative capacity of the network is limited due to a few numbers of positive samples. When $\tau > 70\%$, the overconfidence pseudo labels would easily lead the network to confirmation bias Arazo et al. (2020).We have performed a detailed sensitivity analysis of the model on the given parameters in the section 5.3.

### 4.4 Experimental setup and code

To replicate experiment 1, we utilized the code supplied by the authors. However, certain adjustments were necessary in the data loader function to facilitate the execution of the experiment across all datasets. Details regarding the selection of hyperparameters for each dataset can be found in Table 4.

To conduct the ablation study, we undertook the task of modifying the author's code independently, creating a substantial portion of the functions ourselves.Our contributions to the code can be accessed on our GitHub repository.

The application of the model to non-graph datasets posed the greatest challenges. Since these datasets lacked an adjacency matrix and only contained sets of features and nodes, we addressed this issue by implementing the "construct graph" function based on Bo et al. (2020) methodology. This allowed us to successfully construct the adjacency matrix and implement the model on the non-graph datasets.

Furthermore, during the implementation phase, we observed an issue where the encoder was mapping all nodes into a single point. This resulted in poor K-means clustering performance, leading to model stagnation, especially as it clustered all high-confidence points into a single class. Upon thorough investigation of the codebase, we identified that the construction of smooth features using Laplacian filters, i.e., $\bar{X}$, caused a majority of nodes to share the same feature vector. This, in turn, led to a misencoding of nodes. To address this, when applying the model to non-graph datasets, we opted to bypass the feature smoothing step and instead directly utilized the feature vector for encoding. This adjustment resulted in significantly improved performance.

### 4.5 Computational requirement

All our experiments were conducted on Google Colab, leveraging the T4 GPU. While applying models to graph datasets incurred minimal computational requirements, we encountered a bottleneck when working with non-graph datasets due to limitations in system RAM. To address this challenge, we implemented Principal Component Analysis (PCA) to reduce the dimensionality of the feature vector. This reduction not only mitigated the strain on system resources but also resulted in a decreased computational load.

## 5 Results

In this section, we present the results obtained in our study. In the subsection titled "Experiment 1," we discuss the performance of our model across six datasets, comparing our results with those reported by the authors.

Moving on to the section titled "Ablation Study," we conduct a comprehensive analysis of our model by performing various ablation experiments. Here, we aim to validate the claims outlined in Section 3, specifically focusing on the scope of reproducibility.

Lastly, in the section titled "Results Beyond Paper," we extend the application of our model to additional datasets, both graph and non-graph related. This broader examination allows us to explore the generalizability and versatility of our model beyond the specific datasets originally considered in the paper.

## 5.1 Experiment 1: Performance comparison

In the initial research paper, the authors conducted a comparative analysis of their model against 12 other approaches across 6 datasets namely CORA, CITESEER, UAT, BAT, EAT, & AMAP, details of the datasets are outlined in section 4.2 . In this section, we replicate their results using the code shared by the authors on their GitHub repository. However, we had to develop a custom data loader, as outlined in section 4.4, to adapt the model to the dataset, given the variations in file types.

Based on the data presented in Table 5, which outlines the performance achieved by our approach across

Table 5: Reproduced performance

| Dataset | ACC | NMI | ARI | F1 |
|---|---|---|---|---|
| CORA | 74.13 ± 1.58 | 56.71 ± 1.20 | 51.69 ± 2.50 | 71.01 ± 3.62 |
| CITESEER | 69.95 ± 0.78 | 44.38 ± 0.54 | 44.27 ± 1.44 | 61.48 ± 1.22 |
| AMAP | 77.29 ± 0.62 | 66.74 ± 0.82 | 57.57 ± 1.14 | 71.83 ± 0.90 |
| BAT | 76.72 ± 1.65 | 52.60 ± 1.54 | 50.70 ± 2.43 | 76.08 ± 1.76 |
| EAT | 56.16 ± 1.02 | 32.13 ± 1.11 | 25.65 ± 1.10 | 56.25 ± 0.88 |
| UAT | 54.82 ± 1.33 | 28.55 ± 1.31 | 21.26 ± 2.32 | 54.50 ± 2.03 |

Table 6: Discrepancies with authors'.

| △ACC | △NMI | △ARI | △F1 |
|---|---|---|---|
| 0.24 | 0.26 | -0.82 | 0.03 |
| 0.11 | 0.05 | -1.41 | -1.23 |
| 0.04 | -0.7 | -0.42 | -0.35 |
| 1.68 | 2.37 | 3.75 | 1.18 |
| -1.03 | -1.72 | -2.06 | -0.84 |
| -1.52 | 0.4 | -4.26 | -0.74 |

the six datasets, and Table 6, which highlights the disparities between our results and those obtained by the authors, it can be concluded that our results closely approximate those obtained by the authors.

## 5.2 Ablation study

The authors of the original paper conducted an ablation study, although they did not provide the accompanying code or detailed description. However, leveraging the provided description, we meticulously crafted our own implementation to conduct the study. While the results we obtained may not precisely match those obtained by the author due to the unavailability of code, they do exhibit a similar trend.

Table 7: Performance on graph datasets

| Dataset | metrics | With SCGC & RNS | Only RNS | Add edges | Drop edges | Mask features | Diffusion | CCGCN (reproduced) |
|---|---|---|---|---|---|---|---|---|
| CORA | Accuracy | 46.81 ± 2.75 | 73.70 ± 1.89 | 70.14 ± 1.25 | 71.52± 1.96 | 72.67 ± 1.32 | 64.07±3.84 | **74.13 ± 1.58** |
| | NMI | 26.68 ± 2.60 | **57.18 ± 1.45** | 50.34 ± 1.51 | 53.96 ± 1.65 | 54.57 ± 1.78 | 52.24 ±2.05 | 56.71 ±1.20 |
| | ARI | 17.70 ± 2.66 | 51.36 ± 2.46 | 45.45 ± 1.49 | 48.31 ± 3.08 | 49.5 ± 1.28 | 42.50 ± 4.07 | **51.69 ± 2.50** |
| | F1 | 40.44 ± 2.45 | 68.83 ± 3.59 | 67.40 ± 286 | 68.66 ± 4.05 | 69.26 ± 3.13 | 56.02 ± 4.64 | **71.01 ± 3.62** |
| CITESEER | Accuracy | 63.53± 2.20 | **69.49 ± 0.74** | 65.75 ± 0.58 | 65.75 ± 1.49 | 67.96±1.88 | 68.85 ± 0.97 | 69.95 ± 0.78 |
| | NMI | 36.85± 0.62 | **43.96 ± 0.87** | 36.80 ± 0.59 | 39.51 ± 1.71 | 40.97 ± 1.84 | 42.68 ± 0.94 | 44.38 ± 0.54 |
| | ARI | 34.66 ± 2.35 | **43.90 ± 1.18** | 37.26 ± 0.87 | 37.33 ± 2.73 | 41.69 ± 2.68 | 42.51 ± 1.74 | 44.27± 1.44 |
| | F1 | 57.46 ± 1.30 | **61.59 ± 1.80** | 57.51 ± 0.80 | 57.77 ± 1.13 | 58.93 ±1.75 | 60.35 ± 0.75 | 61.48 ± 1.22 |
| AMAP | Accuracy | 51.09 ± 2.89 | 77.54 ± 0.55 | 69.04 ± 1.17 | 76.57 ± 0.67 | 77.15 ± 0.64 | **77.61 ± 1.49** | 77.29 ± 0.62 |
| | NMI | 33.35 ± 1.82 | 67.01 ± 0.73 | 56.89 ± 1.32 | 66.04 ± 0.86 | 67.02 ± 0.52 | **67.86 ± 1.17** | 66.74 ± 0.82 |
| | ARI | 21.87 ± 1.47 | 58.44 ± 0.84 | 45.83 ± 0.92 | 56.88 ± 0.66 | 57.81 ± 0.86 | **59.09 ± 1.48** | 57.57 ± 1.14 |
| | F1 | 45.81 ± 3.54 | 71.88 ± 0.65 | 66.69 ± 1.92 | **72.04 ± 0.75** | 72.02 ± 0.41 | 71.25 ±3.75 | 71.83 ± 0.90 |
| UAT | Accuracy | 49.46 ± 1.59 | 50.40 ± 1.04 | 51.21 ± 1.69 | 47.07 ± 3.88 | 53.89 ± 1.69 | 47.00 ± 2.67 | **54.82 ± 1.33** |
| | NMI | 21.24 ± 2.92 | 21.72 ± 1.56 | 25.16 ± 2.34 | 22.68 ± 4.23 | 22.68 ± 4.23 | 19.70 ± 2.17 | **28.55 ± 1.31** |
| | ARI | 14.42 ± 2.48 | 14.84 ± 1.47 | 16.96± 2.54 | 14.38 ± 3.59 | **21.31 ± 2.84** | 15.34 ± 2.54 | 21.26±2.32 |
| | F1 | 46.39 ± 2.72 | 47.43 ± 2.91 | 50.72 ± 1.89 | 42.49 ± 5.13 | 53.37 ± 2.05 | 42.25 ± 4.58 | **54.50 ±2.03** |
| EAT | Accuracy | 36.81 ± 8.90 | **56.37 ± 1.20** | 55.08 ± 0.21 | 54.53 ± 0.76 | 55.82 ± 0.75 | 42.43 ± 1.34 | 56.16 ± 1.02 |
| | NMI | 11.40 ± 8.51 | 33.01 ± 1.36 | **34.33± 0.46** | 33.06 ± 1.08 | 32.19 ± 1.17 | 18.77 ± 2.54 | 32.13 ± 1.11 |
| | ARI | 8.32 ± 7.18 | 26.05 ± 1.44 | **28.25 ± 0.29** | 26.28 ± 1.21 | 25.30 ± 0.9 | 12.35 ± 3.30 | 25.65 ± 1.10 |
| | F1 | 27.72 ± 13.59 | 56.10 ± 2.34 | 53.85 ± 0.57 | 50.± 3.137 | 56.03 ± 0.50 | 33.98 ± 2.27 | **56.25±0.88** |
| BAT | Accuracy | 69.38 ± 2.08 | 69.16 ± 21.16 | 64.42 ± 0.85 | 64.58 ± 0.85 | 70.30 ± 3.30 | 48.39 ± 2.29 | **76.72 ± 1.65** |
| | NMI | 47.15 ± 2.29 | 47.65 ± 2.42 | 41.99 ± 1.75 | 49,18± 2.03 | 48.78 ± 2.05 | 30.82 ± 5.31 | **52.60 ± 1.54** |
| | ARI | 40.59 ± 2.45 | 40.54 ± 3.13 | 34.98 ± 1.76 | 43.59 ± 1.89 | 45.79 ± 1.93 | 17.80 ± 5.92 | **50.70± 2.43** |
| | F1 | 68.43 ± 3.29 | 68.22 ± 3.20 | 63.47 ± 1.50 | 59.09 ± 1.77 | 67.74 ±4.53 | 44.08 ± 2.29 | **76.08±1.76** |

### 5.2.1 Effectiveness of the Siamese encoder

We effectively executed the ablation study on the unshared Siamese network by sharing its parameter and subsequently applying diverse graph augmentation techniques mentioned in the original paper, namely, ran-

domly dropping 20% edges ("Drop Edges") Xia et al. (2023), or randomly adding 20% edges ("Add Edges") Xia et al. (2023), or graph diffusion ("Diffusion") with 0.20 teleportation rate Hassani & Khasahmadi (2020), or randomly masking 20% features ("Mask Features") Zhao et al. (2021).

Upon comparing the performance results obtained from these augmentation techniques to the "CCGCN reproduced" performance, it becomes evident that the "CCGCN" performance consistently outperforms the augmented scenarios. This finding strongly supports the authors' claims 2 regarding the superiority of the unshared Siamese encoder over traditional graph augmentation methods. Notably, utilizing the unshared Siamese encoder effectively mitigates semantic drift, further emphasizing its efficacy in preserving the integrity of the encoded information.

### 5.2.2 Effectiveness of DPS

We were unable to replicate the suggested method for assessing the importance of DPS suggested in the original paper due to the unavailability of code. Instead, to evaluate the effectiveness of DPS, we substituted DPS with the neighbor-oriented contrastive loss method proposed in SCGC by Liu et al. (2022). By doing so, we essentially removed the DPS component from our experiment

Upon comparing the results, it became evident that the CCGCN(reproduced) performance, achieved by replacing DPS with the neighbor-oriented contrastive loss, under performed compared to the original results. This comparison clearly indicates that the introduction of DPS significantly enhances the quality of positive samples, resulting in improved performance in graph clustering tasks.

In conclusion, our findings support the claim made by the authors that the DPS method enhances the quality of positive samples, leading to improved performance in graph clustering tasks.

### 5.2.3 Effectiveness of RNS

We encountered similar difficulties in replicating the suggested method for evaluating the importance of Random Negative Sampling (RNS) due to the absence of available code. However, to validate claim 3, we opted to assess the performance of the model solely on RNS-generated negative samples. This approach allowed us to directly evaluate the effectiveness of the proposed RNS methodology

Upon comparing the results obtained from this experiment with the CCGCN(reproduced) performance, we observed that the performance of the model when trained exclusively on RNS-generated negative samples closely resembled the overall performance. This similarity in performance strongly suggests that RNS is proficient in generating reliable negative samples. Consequently, these negative samples enhance the informativeness of the training data, thereby improving the model's capacity to differentiate between various graph clusters.

In summary, our findings support claim 3 by demonstrating that RNS is indeed capable of generating reliable negative samples, which in turn enhances the effectiveness of the training data and improves the model's ability to discern between different graph clusters.

### 5.3 Hyperparameter sensitivity

We conducted a study to assess hyperparameter sensitivity (fig 2) in our research. Specifically, we fixed the value of $\alpha$ and varied $\tau$ to examine the sensitivity of $\tau$. Conversely, we fixed the value of $\tau$ and varied $\alpha$ to investigate the sensitivity of $\alpha$.

To conduct this study, we initially set $\alpha$ to 0.5 and adjusted $\tau$ within the range of 0.5 to 0.7. This range was chosen based on the findings that when $\tau$ falls below 50%, the network's discriminative capacity becomes limited, and when $\tau$ exceeds 70%, the pseudo labels' overconfidence may introduce confirmation bias Arazo et al. (2020). Following this, we maintained $\tau$ at 0.5 and varied $\alpha$ from 0.01 to 10.

For all values of $\alpha$ except 10, the approach was successfully executed on the CITESEER dataset. However, at alpha 10, the approach exceeded the available GPU RAM. The specific ranges and standard deviations for each dataset are outlined in Tables 8 and 9

Table 8: Ranges of metrics for sensitivity analysis

| Dataset | Sensitivity on $\tau$ | | | | Sensitivity of $\alpha$ | | | |
|---|---|---|---|---|---|---|---|---|
| | ACC | NMI | ARI | F1 | ACC | NMI | ARI | F1 |
| CORA | 0.7533 | 1.5002 | 0.6856 | 1.4070 | 0.6869 | 0.7565 | 0.8443 | 1.6598 |
| CITESEER | 0.5260 | 0.5391 | 0.6180 | 0.4081 | 0.0060 | 0.1782 | 0.1007 | 0.5528 |
| AMAP | 0.2471 | 0.0483 | 0.6675 | 0.3572 | 0.1817 | 0.2514 | 0.5425 | 0.2027 |
| UAT | 1.0840 | 0.4384 | 0.6859 | 1.0695 | 1.0672 | 1.3809 | 1.6853 | 2.6035 |
| EAT | 0.4010 | 0.5605 | 0.5253 | 0.4433 | 0.4511 | 0.3035 | 0.2251 | 0.4127 |
| BAT | 0.7634 | 0.9180 | 0.6498 | 0.9138 | 0.3817 | 0.6585 | 0.3421 | 0.7325 |

Table 9: Standard deviations of metrics for sensitivity analysis

| Dataset | Sensitivity on $\tau$ | | | | Sensitivity of $\alpha$ | | | |
|---|---|---|---|---|---|---|---|---|
| | ACC | NMI | ARI | f1 score | ACC | NMI | ARI | f1 score |
| CORA | 0.4126 | 0.7554 | 0.3497 | 0.7071 | 0.2928 | 0.3167 | 0.3505 | 0.7251 |
| CITESEER | 0.2946 | 0.2802 | 0.3462 | 0.2133 | 0.0035 | 0.1029 | 0.0581 | 0.3192 |
| AMAP | 0.2471 | 0.0483 | 0.6675 | 0.3572 | 0.1817 | 0.2514 | 0.5425 | 0.2027 |
| UAT | 0.5687 | 0.2442 | 0.3747 | 0.5469 | 0.5031 | 0.5751 | 0.8374 | 1.2408 |
| EAT | 0.2026 | 0.2914 | 0.2958 | 0.2339 | 0.0979 | 0.1189 | 0.2558 | 0.0995 |
| BAT | 0.3842 | 0.4590 | 0.3278 | 0.4598 | 0.2067 | 0.3141 | 0.1639 | 0.4085 |

Through an examination of hyperparameter sensitivity across six datasets, it becomes evident that while adjustments to hyperparameters may result in slight enhancements, the model demonstrates remarkable resilience and stability in performance. Despite potential variations in hyperparameter settings, the overall impact on model performance remains minimal.

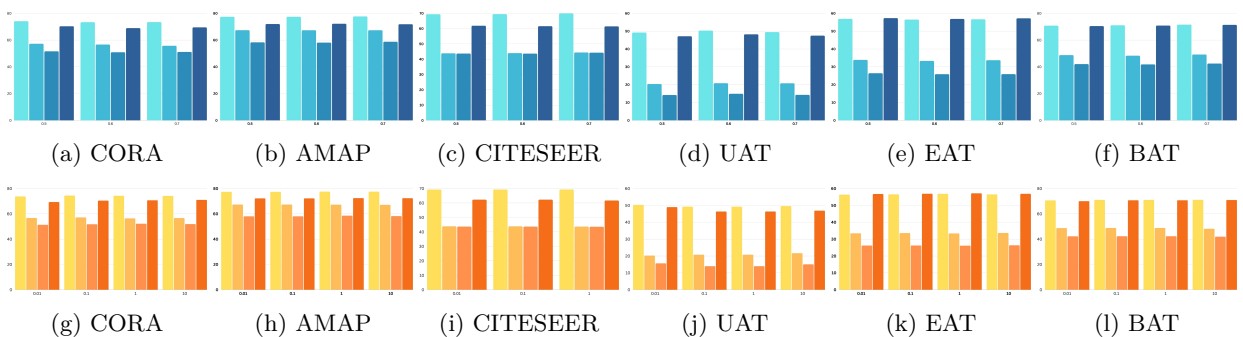

(a) CORA      (b) AMAP      (c) CITESEER      (d) UAT      (e) EAT      (f) BAT

(g) CORA      (h) AMAP      (i) CITESEER      (j) UAT      (k) EAT      (l) BAT

Figure 1: Sensitivity study on $\tau$(top) and $\alpha$(bottom)

# 6 Results beyond original paper

### 6.0.1 Non graph datasets

In our study extension, our primary focus was on testing the generalizability of the models. To achieve this, we utilized the model for node classification on non-graph datasets, specifically the USPS and REUT datasets. The evaluation of the models on these non-graph datasets revealed some important findings.

Firstly, despite demonstrating reasonable performance on non-graph datasets, we identified several shortcomings. One significant issue was the high computational load associated with our approach.

### 6.0.2 Computational challenges and solutions

In both the USPS and REUT datasets, we encountered a significant computational burden, notably in terms of GPU RAM, which exceeded 15 GB. This heightened demand primarily stemmed from the k-means clustering step within the model.

**CPU/GPU Trade-off in K-means Clustering Computations**: In k-means clustering, a strategic trade-off can be made during distance calculations (while calculating dis = (A-B)**2 ). By default, these calculations take place in GPU memory, but an alternative using CPU memory is possible which we have applied while performing Cross Validation on these datasets. However, it's important to note that such a switch extends the overall training time.

**Precision Trade-off**: In k-means clustering, a strategic trade-off can be made during distance calculations (while calculating dis = (A-B)**2 ) converting these Matrices to less precise float values (i.e A.float() and B.float()) introduces a tradeoff between precision and computational efficiency doing this may reduce model's performance but allows faster and memory efficient computations

**Dimensionality Reduction**:To address this issue, we propose employing Principal Component Analysis (PCA) to diminish the dimensionality of the feature vector. Our analysis revealed that PCA has a minimal impact on model performance, yet it substantially alleviates the computational load.
In summary, integrating PCA into our workflow offers a viable solution to mitigate the computational demands associated with k-means clustering, without compromising model effectiveness.

### 6.0.3 Model stagnation and solution

Secondly, we encountered poor results when attempting to employ Laplacian filters for feature smoothing. As detailed in Section 4.4, our experiments revealed that applying Laplacian filters for feature smoothing resulted in all nodes having identical features. Consequently, this led to misencoding by the unshared encoder, where the

Table 10: Performance on non-graph datasets

| Dataset | ACC (%) | NMI (%) | ARI (%) | F1 (%) |
|---|---|---|---|---|
| REUT | 74.34 ± 2.18 | 53.76 ± 2.89 | 51.92 ± 4.34 | 60.86 ± 1.59 |
| USPS | 73.68 ±3.23 | 63.97 ±1.55 | 58.08 ±2.54 | 72.33±3.79 |

majority of nodes were encoded similarly. This, in turn, compromised the construction of pseudo labels and subsequently, the overall model performance. By bypassing the feature smoothing step in our experiments, we achieved significantly improved results, as outlined in Table 9.

In summary, while the method demonstrated promise for node classification on non-graph datasets, we encountered challenges related to high computational demands and ineffective feature smoothing techniques. These findings underscore the importance of refining our approach to address these limitations for more robust model performance

### 6.0.4 Graph datasets

Table 11: Performance on Additional graph datasets

| Dataset | ACC | NMI | ARI | F1 |
|---|---|---|---|---|
| ACM | 89.51 ± 0.55 | 66.01 ± 1.21 | 71.68 ± 1.29 | 89.46 ± 0.57 |
| DBLP | 52.82 ± 2.29 | 24.04 ± 2.48 | 17.85 ± 1.91 | 53.28 ± 2.15 |
| AMAC | 55.70 ± 1.18 | 34.92 ± 1.62 | 31.38 ± 1.25 | 35.71 ± 4.25 |
| TEXAS | 47.48 ± 0.62 | 14.26 ± 0.96 | 12.46 ± 1.52 | 31.27 ± 1.91 |
| WISC | 49.72 ± 5.05 | 24.90 ± 7.12 | 11.82 ± 7.77 | 30.73 ± 6.41 |
| PUBMED | 63.09 ± 3.76 | 28.54 ± 2.81 | 26.57 ± 3.03 | 61.92 ± 4.70 |

We further extended our approach to other graph datasets with the aim of studying the robustness of the model across datasets of various sizes. The results of this extension are summarized in Table 11

Despite our thorough analysis, we have been unable to discern any clear patterns regarding the varying performance of the graph datasets. Notably, TEXAS and WISC, despite being relatively small datasets, exhibit poor performance when evaluated with the model. Conversely, ACM, which is of slightly larger when compared to TEXAS and WISC, demonstrates remarkably high performance. Additionally, the model achieves decent performance on the relatively large dataset PUBMED.
This inconsistency in performance across datasets suggests that factors beyond dataset size may significantly influence the model's effectiveness. Further investigation is required to identify these factors and understand their impact on the model's performance across different graph datasets.
We further conducted a hyperparameter sensitivity study on additional datasets. The results indicate that these datasets follow the same trend as previously mentioned, with the model showing minimal changes

in performance with variations in hyperparameters. We observed that on these datasets, that the model showed no performance change on variation with alpha thus those tables are not mentioned. The results of this experiments can be found on the mentioned GitHub link.

Table 12: Standard deviation of metrics for sensitivity analysis

| Dataset | Sensitivity on $\tau$ | | | |
|---|---|---|---|---|
| | ACC | NMI | ARI | F1 |
| ACM | 0.082 | 0.258 | 0.224 | 0.079 |
| DBLP | 0.258 | 0.262 | 0.396 | 0.351 |
| TEXAS | 0.517 | 0.045 | 0.383 | 1.191 |
| WISC | 0.043 | 0.035 | 0.059 | 0.085 |
| AMAC | 0.041 | 0.234 | 0.166 | 0.667 |

Table 13: Ranges of metrics for sensitivity analysis

| Dataset | Sensitivity on $\tau$ | | | |
|---|---|---|---|---|
| | ACC | NMI | ARI | F1 Score |
| ACM | 0.15 | 0.49 | 0.44 | 0.14 |
| DBLP | 0.51 | 1.37 | 0.78 | 0.69 |
| TEXAS | 1.93 | 0.95 | 1.69 | 2.27 |
| WISC | 0.08 | 0.07 | 0.11 | 0.14 |
| AMAC | 0.09 | 0.57 | 0.34 | 1.24 |

## 6.1 Model cross validation

We expanded upon the findings of the original paper by conducting a 10 fold random cross-validation study to assess the performance of the model. The results of this study are outlined below.

| Dataset | metrics | 1 | 2 | 3 | 4 | 5 | 6 | 7 | 8 | 9 | 10 | Average | STD |
|---|---|---|---|---|---|---|---|---|---|---|---|---|---|
| **Graph Datasets** | | | | | | | | | | | | | |
| BAT | Test Accuracy | 70.37 | 71.43 | 58.82 | 57.14 | 63.64 | 88.89 | 71.43 | 60.00 | 75.00 | 75.00 | 69.172 | 9.137967827 |
| | Test NMI | 52.77 | 54.89 | 46.49 | 58.52 | 68.89 | 85.28 | 61.17 | 65.16 | 66.67 | 85.71 | 64.555 | 12.27686137 |
| | Test ARI | 43.87 | 38.00 | 19.92 | 30.07 | 46.03 | 76.92 | 16.00 | 15.38 | 0.00 | 0.00 | 28.619 | 22.37656875 |
| | Test F1 Score | 56.80 | 66.64 | 58.28 | 50.95 | 53.17 | 86.67 | 71.11 | 61.11 | 73.33 | 77.78 | 69.172 | 10.97177852 |
| EAT | Test Accuracy | 52.5 | 50 | 45.1 | 56.1 | 57.58 | 61.54 | 42.86 | 64.71 | 71.43 | 72.73 | 67.89 | 5.68 |
| | Test NMI | 33.01 | 41.51 | 29.44 | 53.75 | 31.73 | 47.62 | 23.14 | 54.93 | 59.35 | 69.52 | 61.63 | 4.48 |
| | Test ARI | 22.96 | 28.49 | 14.96 | 38 .92 | 15.23 | 31.06 | 1.8 | 36.07 | 44.32 | 39.56 | 47.54 | 6.22 |
| | Test F1 Score | 52.15 | 47.86 | 44.92 | 42.74 | 60.6 | 56.13 | 44.41 | 51.94 | 66.18 | 70.6 | 62.10 | 7.36 |
| AMAP | Test Accuracy | 5.71 | 77.29 | 66.12 | 6964 | 73.68 | 62.87 | 72.82 | 61.99 | 66.54 | 70.24 | 67.89 | 5.68 |
| | Test NMI | 53.15 | 65.16 | 58.47 | 62.46 | 64.74 | 55.79 | 64.04 | 59.45 | 65.15 | 67.94 | 61.63 | 4.48 |
| | Test ARI | 38 | 56.86 | 41.46 | 49.86 | 54.55 | 41.74 | 52.11 | 40.61 | 49.24 | 51.02 | 47.54 | 6.22 |
| | Test F1 Score | 50.6 | 71.75 | 66.76 | 69.12 | 68.6 | 55.53 | 63.78 | 51.87 | 56.36 | 66.67 | 62.10 | 7.36 |
| CITSEER | Test Accuracy | 40.24 | 41.28 | 45.54 | 42.82 | 52,01 | 44.04 | 59.2 | 40. 71 | 36.61 | 46.07 | 44.852 | 6.18 |
| | Test NMI | 22. 47 | 23.15 | 25.29 | 25.57 | 33. 96 | 27.788 | 38.06 | 20.557 | 27.32 | 35.35 | 27.96 | 5.61 |
| | Test ARI | 11.28 | 13.39 | 18.15 | 15.16 | 22.38 | 18.65 | 28.86 | 10.64 | 6.67 | 20.01 | 16.51 | 6.13 |
| | Test F1 Score | 30.18 | 33.76 | 40.47 | 25.93 | 42.35 | 34.96 | 46.04 | 32.90 | 30.79 | 40.7 | 36.808 | 5.03 |
| UAT | Test Accuracy | 33.19 | 47.12 | 50.33 | 40.98 | 51.02 | 34.62 | 48.39 | 46 | 45 | 37.5 | 43.41 | 6.13 |
| | Test NMI | 8.94 | 24.25 | 18.53 | 12.58 | 31.42 | 15.85 | 25.47 | 17.26 | 22.37 | 19.44 | 19.615 | 6.21 |
| | Test ARI | 1.81 | 9.87 | 13.96 | 4.03 | 16.32 | 1.57 | 10.79 | 4.99 | 9.1 | 3.1 | 7.55 | 4.94 |
| | Test F1 Score | 24.53 | 43.78 | 40.75 | 39.57 | 46.42 | 34.78 | 45.57 | 18.88 | 43.85 | 33.79 | 39.19 | 6.34 |
| CORA | Test Accuracy | 53.61 | 53.92 | 49.86 | 51.99 | 47.75 | 51.12 | 56.34 | 55.26 | 56.o4 | 52.05 | 52.89 | 2.67 |
| | Test NMI | 44.64 | 46.77 | 45.07 | 47.86 | 44.58 | 44.88 | 47.72 | 46.71 | 48.84 | 52.37 | 46.94 | 2.30 |
| | Test ARI | 28.17 | 32.37 | 30.73 | 30.02 | 24.70 | 25.41 | 31.89 | 32.07 | 31.71 | 33.18 | 30.02 | 2.82 |
| | Test F1 Score | 47.12 | 45.80 | 36.26 | 41.73 | 44.73 | 45.79 | 39.32 | 42.31 | 47.44 | 37.73 | 42.82 | 3.79 |
| **Non - Graph Datasets** | | | | | | | | | | | | | |
| USPS | Test Accuracy | 72.07 | 59.63 | 71.03 | 74.4 | 75.74 | 71.02 | 68.54 | 57.39 | 71.77 | 69.38 | 68.76 | 5.67 |
| | Test NMI | 63.69 | 64.24 | 72.1 | 65.6 | 79.38 | 64.31 | 58.66 | 56.19 | 66.11 | 78.45 | 67.22 | 7.24 |
| | Test ARI | 57.1 | 46.05 | 57.41 | 60.87 | 63.85 | 57.19 | 49.78 | 49.62 | 56.5 | 56.5 | 55.30 | 5.15 |
| | Test F1 Score | 71.24 | 71.47 | 70.66 | 73.66 | 74.71 | 71.19 | 71.06 | 71.8 | 74.66 | 72.47 | 72.40 | 1.44 |
| REUT | Test Accuracy | 61.1 | 76.56 | 65.16 | 68.75 | 56.34 | 73.63 | 64.69 | 62.86 | 61.01 | 60.07 | 65.017 | 5.97 |
| | Test NMI | 39.09 | 50.29 | 49 | 47.24 | 34.16 | 51.05 | 50.19 | 41.66 | 36.2 | 34.61 | 43.349 | 6.58 |
| | Test ARI | 35.71 | 57.29 | 44.85 | 44.11 | 25.36 | 46.84 | 43.16 | 32.16 | 30.43 | 31.55 | 39.14 | 9.19 |
| | Test F1 Score | 56.01 | 67.91 | 61.97 | 62.05 | 55.95 | 61.24 | 56.93 | 56.87 | 59.96 | 50.43 | 58.932 | 4.51 |

Table 14: Results of Experiments

**Low Variance in Evaluation Metrics:** The evaluation metrics for the test set show minimal deviations when compared to those of the training set. This indicates that the model performs consistently well on unseen data, suggesting low variance. The model's capacity to accurately classify unseen nodes within the test graph dataset demonstrates its robustness.

**Unusual Behavior in Smaller Datasets:** In smaller datasets, we observe a peculiar trend where evaluation metrics display higher standard deviations from the mean values. This increased variability can be

attributed to the limited size of the test set. With fewer instances available for evaluation, the metrics may fluctuate significantly. This underscores the importance of taking into account dataset size and ensuring a sufficiently large test set for accurate performance assessment.

In summary, the consistent evaluation metrics observed for the test set indicate the model's ability to generalize effectively to unseen data. Nevertheless, particular attention is warranted for smaller datasets, where fluctuations in metrics may occur due to the limited size of the test set. This underscores the importance of carefully considering dataset characteristics and employing appropriate experimental design to ensure reliable model evaluation and comparison.

## 7 Discussion

### 7.1 verification of claim

In this section, we provide a summary of whether our results support the original claims made by the authors. The validation for all claims has been conducted in Section 5.2, Ablation Study.In this section we summarize weather our results support the original claims made by the authors. The validation to all the claims have been performed in the section 5.2, Ablation study.

- Claim 1 states the effectiveness of Deep Positive Sampling (DPS) and emphasizes the importance of well-discriminative positive samples. In Section 5.2.2, we address this claim by replacing the positive contrastive loss with the neighborhood contrastive loss, as utilized in SCGC. This substitution effectively removes DPS from our model. By comparing the performance obtained with this alteration to that achieved with the total loss, we observe superior performance when employing DPS. As a result, we validate claim 1.

- Claim 2 asserts the effectiveness of the Unshared Siamese network compared to traditional graph augmentation methods by avoiding semantic drift. To validate this claim, we initially share the parameters of the Siamese encoder and subsequently apply various graph augmentation techniques as outlined in Section 5.2.2. We then compare the performance obtained from these augmented methods to that of the actual method.
  Through this comparison, we observe that in the majority of cases, the performance of the actual method surpasses that of the augmented methods. This observation leads us to the conclusion that unshared Siamese encoders are superior in avoiding semantic drift, thereby validating claim 2.

- Claim 3 highlights the significance of Random Negative Sampling (RNS) in constructing more reliable negative samples, thereby contributing significantly to the learning process. While we were unable to replace RNS with regular negative loss as suggested by the authors to test the effectiveness of RNS, we validated claim 3 by utilizing only negative contrastive loss. By removing the positive loss component and comparing the results obtained with this approach to those obtained using the total loss, we observed a close similarity between the two sets of results. This observation strongly supports the important contribution of RNS, as the performance achieved solely on RNS closely approached that obtained with the total loss. Hence, claim 3 has been validated.

- **Our findings and contributions :** Based on the extensive experiments conducted, our findings reveal that the model exhibits a broad spectrum of performance and lacks consistency. This variability in performance is not solely attributable to differences in dataset size. Additionally, we demonstrated the model shows negligible changes to performance on variation with hyper parameter $\alpha$ and $\tau$, In some cases showing 0 changes.
  Our primary contribution lies in successfully applying the technique to non-graph datasets. Despite encountering some limitations, we thoroughly discussed proposed solutions to address these challenges throughout our study.
  Lastly we also conducted a cross validation study which was absent in the original paper, and were able to Low Variance in Evaluation Metrics & some unusual Behavior in Smaller Datasets which are discussed above in detail.

### 7.2   Overall conclusion

In summary, our study successfully validated the claims made by the authors and reproduced the model's performance as reported in the original paper. However, the results from our ablation study did not precisely match those obtained by the authors. Although the results of other experiments in the ablation study yielded lower performance, their proximity to the results of CCGCN is concerning. Expanding our investigation to non-graph datasets posed initial challenges due to computational load, which we mitigated by employing PCA on the feature matrix. Additionally, we encountered issues with feature smoothing, resulting in poor encoder performance. Once we bypassed these obstacle, our model achieved commendable results
Furthermore, upon applying our model to additional graph datasets, we noted inconsistencies in performance across datasets. This observation suggests that factors beyond dataset size significantly influence the model's effectiveness.
In conclusion, while our study confirmed the validity of the authors' claims and successfully replicated their reported performance, it also revealed areas for further exploration and improvement, particularly concerning the model's response to different dataset characteristics.

### 7.3   What was easy

The code provided by the authors was extensive and relied on open-source standard libraries, which helped ensure reduced implementation errors. Additionally, the methodology described in the paper was comprehensive and easy to understand.

### 7.4   What was difficult

We encountered significant difficulties during the ablation study due to the unavailability of code. Additionally, applying the model to non-graph datasets proved challenging, primarily because of the high computational requirements of the task.

### 7.5   Communication with authors

We attempted to contact the authors regarding the code for the ablation study but have not received any response till date.

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
