# OpenReview forum: "[Re]Reproducibility Study: Cluster-guided Contrastive Graph Clustering Network"
_TMLR — Rejected by TMLR_

### Review · Reviewer_dqLC · 2024-02-23

**Summary Of Contributions:**

This research is dedicated to evaluating the reproducibility of the study "Cluster-guided Contrastive Graph Clustering Network" by Yang et al. (2023). The objective is to substantiate the assertions presented within the publication. The claims under scrutiny include:
1. Introduction of a Discriminative Positive Sample Extraction Mechanism (DPS): The paper posits that the newly introduced DPS mechanism exhibits enhanced discriminative capabilities.
2. Development of an Unshared Siamese Encoder to Mitigate Semantic Drift: The authors advocate for an innovative unshared Siamese encoder design to avert the potential for semantic drift resulting from improper graph augmentation.
3. Creation of an Algorithm for Assembling Reliable Negative Samples (RNS): The study advances a novel algorithm intended to generate more dependable negative samples, termed as RNS.

**Audience:**

No

**Claims And Evidence:**

Yes

**Requested Changes:**

1. Does the study uncover any noteworthy conclusions or
2. Does the study provide any illuminating perspectives that advance the current understanding?

**Strengths And Weaknesses:**

Strengths:
The study presents a thorough evaluation of the Cluster-guided Contrastive Graph Clustering Network (CCGCN), encompassing a performance comparison with ablation studies, and extends its application to additional datasets.

Weaknesses:
1. The present study primarily serves to verify the existing work, with its scope confined to assessing the reproducibility of the original findings. Consequently, it does not yield novel insights that diverge from the initial conclusions drawn in the original research.
2. As a validation study, it remains ambiguous whether the methodology incorporates cross-validation procedures with varied random seeds. Such measures are crucial to ensure a fair comparison and robust verification of results. For guidance on best practices in this area, one might consult the study "A Fair Comparison of Graph Neural Networks for Graph Classification," which exemplifies the implementation of these techniques.

---

> ### Author Response · Authors · 2024-03-09
> **Reply to Requested changes and weaknesses**
>
> We  thank you for the reviews and suggestions to our study.
>
> Based on your suggestion we have implemented a cross validation study on the model and mentioned the results and findings under section 6.1 " Model cross-validation".
> The findings and conclusion to our study are as follows, \
> Our experiments reveal that the model's performance exhibits considerable variability, which cannot be solely attributed to differences in dataset size. Furthermore, we demonstrated that the model displays minimal changes in performance even with variations in hyperparameters α and τ, sometimes showing no changes at all.
>
> Our primary contribution stems from successfully applying the technique to non-graph datasets. Despite encountering limitations, we extensively discussed proposed solutions to overcome these challenges throughout our study. Additionally, we conducted a cross-validation study, which was absent in the original paper, thereby enhancing the robustness of our findings.

---

> ### Comment · Action_Editor_rX42 · 2024-03-22
> **Please make the final recommendation**
>
> Thanks for reviewing the TMLR submission.
> Please just finish the last step of your review as soon as possible, thanks.

---

> ### Comment · Action_Editor_rX42 · 2024-03-28
> **Warning: Please make the final recommmendation immedieately!**
>
> TMLR submission needs your recommendation before we can proceed to the next step.

---

> ### Comment · Action_Editor_rX42 · 2024-04-06
> **Warning Again: Please make the final recommmendation immedieately!**
>
> TMLR submission needs your recommendation RIGHT NOW before we can proceed to the next step.

---

### Review · Reviewer_g1Lx · 2024-03-03

**Summary Of Contributions:**

The paper presents a reproducibility study of the Cluster-guided Contrastive Graph Clustering Network (CCGCN), focusing on validating the original claims and assessing the model's generalizability. It expands the evaluation to 12 graph and 2 non-graph datasets, critically examining the Discriminative Positive Sample Extraction Mechanism (DPS), the unshared Siamese encoder to prevent Semantic Drift, and the Reliable Negative Sample (RNS) construction. The study also explores the computational challenges and effects of hyperparameters on model performance.

**Audience:**

No

**Broader Impact Concerns:**

NA.

**Claims And Evidence:**

Yes

**Requested Changes:**

Here are some suggestions for improving the paper:
- Provide a more detailed comparison between the original findings and the reproducibility study outcomes, specifically highlighting the discrepancies in performance metrics.
- Address the computational challenges and suggest optimizations for applying the model to non-graph datasets.
- Explore additional techniques to mitigate semantic drift beyond the unshared Siamese encoder, providing a broader context for the method's effectiveness.
- Include a more detailed sensitivity analysis of hyperparameters, particularly focusing on their impact on model generalizability across diverse datasets.
- Attempt further outreach to the original authors for comments or insights, which could enrich the reproducibility study's findings.

**Strengths And Weaknesses:**

Strengths:
- Comprehensive reproducibility scope, including additional datasets beyond the original paper.
- Detailed examination of proposed mechanisms (DPS, Siamese encoder, RNS) through ablation studies.
- Clear communication of methodological adjustments and their rationale.

Weaknesses:
- Limited novelty and contributions.
- Encountered challenges in replicating exact experimental setups due to computational limitations and unavailability of complete original code.
- Difficulties in applying the model to non-graph datasets, indicating a potential limitation in generalizability.

---

> ### Author Response · Authors · 2024-03-09
> **Reply to requested changes**
>
> Thank you for reviews and suggestions to our study.  We have taken into consideration the requested changes and our reply is as follows.
>
> Req 1.  Detailed comparison between the original findings and the reproducibility study outcomes: \
> A table showing the difference between the results of author and our study is mentioned in section 5.1. It can clearly be seen that the differences are small.
>
> Req 2.  Address the computational challenges and suggest optimizations for applying the model to non-graph datasets: \
> A section under non graph dataset has been added highlighting the problem and solution.
>
> Req 3. Explore additional techniques to mitigate semantic drift beyond the unshared Siamese encoder: \
> We were not able to come up with technique other than Siamese encoder, we have contacted the original authors for the same, but have received no response. We tired using autoencoder but we faced many challenges and were not able to show any significant results.
>
> Req 4. Include a more detailed sensitivity analysis of hyperparameters, particularly focusing on their impact on model generalizability across diverse datasets: \
> Hyperparameter sensitivity analysis were conducted on the additional datasets and its results were mentioned, showing the same outcome that model shows minimal changes to hyperparameter changes. In some cases showing 0 change. The results are mentioned under the section of " additional graph datasets"

---

### Review · Reviewer_8aeD · 2024-03-05

**Summary Of Contributions:**

This paper is a reproducibility of the paper 'Cluster-guided Contrastive Graph Clustering Network', which mainly focuses on improving the quality of positive and negative samples in graph contrastive learning by taking the clustering information into considering.

The main conclusion of this work is most claims made by the authors of 'Cluster-guided Contrastive Graph Clustering Network' are sucessfully validated, except that the ablation study results do not match precisely. The authors also extend to test the method on more graph datasets and non-graph datasets.

**Audience:**

Yes

**Claims And Evidence:**

No

**Requested Changes:**

Maybe the code and the additional datasets used in reproducing the work should be released to make the study more convincing.

**Strengths And Weaknesses:**

Strengths:

This paper provides a thorough check on the reproducibility of the paper 'Cluster-guided Contrastive Graph Clustering Network', and also perform further test on more datasets to check the capability of the method proposed in 'Cluster-guided Contrastive Graph Clustering Network'.

Weakness:

Although checking the reproducibility is an important perspective of scientific research, the results are more like a technical report and therefore seems not suitable to appear in TMLR, which focus on developing new machine learning techniques. Moreover,

---

> ### Author Response · Authors · 2024-03-05
> **Regrading code and additional datasets**
>
> We greatly appreciate your review and suggestions for changes. We want to bring to your attention to the fact that we have attached the link to the anonymous GitHub as a footnote in our paper, which contains the code for all experiments conducted in the paper. More over we have also mentioned the source for all the datasets in our paper. For your convenience we have mentioned both the links below again.
>
> Our implementation : https://anonymous.4open.science/r/Rep_CCGC-BB93/README.md
> Datasets : https://github.com/yueliu1999/Awesome-Deep-Graph-Clustering

---

> > ### Comment · Reviewer_8aeD · 2024-03-20
> > **Thanks for clarifying the concern regarding code and datasets**
> >
> > Thanks for clarifying the concern regarding code and datasets, this concern is resolved.

---

### Decision · Action_Editor_rX42 · 2024-04-08

**Recommendation:** Reject

**Comment:**

All reviewers recommended rejection. Concretely, the reviewers pointed out that the work encounters "challenges in replicating exact experimental setups", has "a potential limitation in generalizability", and the validation settings "remains ambiguous". After seeing the responses of the authors, all the reviewers still hold negative scores. The AE have read the paper and agree that this work has significant limitations. The decision of AE is made based on this consensus.

**Audience:**

This paper primarily targets audiences that focus on graph-based learning. However, the findings could be less interesting because the conclusions of this work are vague (e.g., factors beyond dataset size significantly influence the model’s effectiveness), and the experimental settings are tricky.

**Claims And Evidence:**

Not well supported. This work reproduces the work CCGCN and claims that its performance lacks consistency. However, as stated in the paper, the authors can not discern any clear patterns regarding the varying performance. Thus, the reviewers point out that the experimental setting may be not well set, which affects the reliability of the results and thus cannot support the claim.